# Cholesterol and Egg Intakes, and Risk of Hypertension in a Large Prospective Cohort of French Women

**DOI:** 10.3390/nu12051350

**Published:** 2020-05-08

**Authors:** Conor-James MacDonald, Anne-Laure Madika, Fabrice Bonnet, Guy Fagherazzi, Martin Lajous, Marie-Christine Boutron-Ruault

**Affiliations:** 1INSERM (Institut National de la Santé et de la Recherche Médicale) U1018, Center for Research in Epidemiology and Population Health (CESP), Institut Gustave Roussy, 94805 Villejuif, France; conor.macdonald@gustaveroussy.fr (C.-J.M.); Annelaure.MADIKA@CHRU-LILLE.FR (A.-L.M.); fabrice.bonnet@chu-rennes.fr (F.B.); guy.fagherazzi@gmail.com (G.F.); 2Université Paris-Saclay, Université Paris-Sud, 94805 Villejuif, France; 3Université de Lille, CHU Lille, EA 2694-Santé publique: épidémiologie et qualité des soins, F-59000 Lille, France; 4Faculty of Medicine, Université Rennes1, F-35043 Rennes, France; 5Department of endocrinology diabetes and nutrition, CHU Rennes, F-35033 Rennes, France; 6Department of Population Health, Luxembourg Institute of Health, L-1445 Strassen, Luxembourg; 7Center for Research on Population Health, INSP (Instituto Nacional de Salud Pública), Cuernavaca 62100, Mexico; mlajous@insp.mx; 8Department of Global Health and Population, Harvard T.H. Chan School of Public Health, Boston, MA 02115, USA

**Keywords:** eggs, hypertension, cholesterol, epidemiology, prospective studies

## Abstract

**Purpose:** The relationship between egg and cholesterol intakes, and cardiovascular disease is controversial. Meta-analyses indicate that egg consumption is associated with increased cardiovascular disease and mortality, but reduced incidence of hypertension, a major risk factor for cardiovascular disease. This study aims to investigate the associations between consumption of egg and cholesterol, and hypertension risk in a cohort of French women. **Methods:** We used data from the E3N cohort study, a French prospective population-based study initiated in 1990. From the women in the study, we included those who completed a detailed diet history questionnaire, and who did not have prevalent hypertension or cardiovascular disease at baseline, resulting in 46,424 women. Hypertension cases were self-reported. Egg and cholesterol intake was estimated from dietary history questionnaires. Cox proportional hazard models with time-updated exposures were used to calculate hazard ratios. Spline regression was used to determine any dose–respondent relationship. **Results:** During 885,321 person years, 13,161 cases of incident hypertension were identified. Higher cholesterol consumption was associated with an increased risk of hypertension: HR_Q1–Q5_ = 1.22 [1.14:1.30], with associations similar regarding egg consumption up to seven eggs per week: HR_4–7 eggs_ = 1.14 [1.06:1.18]. Evidence for a non-linear relationship between hypertension and cholesterol intake was observed. **Conclusions:** Egg and cholesterol intakes were associated with a higher risk of hypertension in French women. These results merit further investigation in other populations.

## 1. Introduction

The relationship between cholesterol or egg consumption and the onset of cardiovascular diseases (CVD) is unclear, with observational studies and trials finding inconsistent results [1,2,3,4]. Eggs are a main source of dietary cholesterol, and their consumption has been associated with an increased risk of cardiovascular disease [5], all-cause mortality [6], haemorrhagic stroke [7], and diabetes [8,9]; paradoxically, in meta-analysis eggs are associated with a decreased risk of hypertension [10], which is a major risk-factor for cardiovascular disease [11]. Further, results are inconsistent between regions, and results are conflicting when studying specific CVD such as stroke or ischemic heart disease [12,13,14,15,16,17,18,19,20]. Aside from cholesterol, eggs are a source of essential nutrients, vitamins, and high-quality proteins [21], which may be a reason for the inconsistencies, as overall diet quality may be important in the relationship between eggs and CVD.

It has been demonstrated that higher dietary cholesterol can raise serum cholesterol [22], which is in turn associated with blood pressure [23]. High levels of dietary cholesterol have been associated with increased systolic blood pressure in the INTERMAP study [24] conducted in China, Japan, UK and USA, but not in the Finnish Kuopio Ischaemic Heart Disease Risk Factor Study (KIHD) [17]. Recently, Mazidi et al. observed increased systolic and diastolic blood pressure in higher consumers of eggs, but found no relationship with cardiovascular mortality [25]. The complexity of the association between egg/cholesterol intake and CVD is reflected in confusing guidelines for egg and cholesterol consumption. As of 2015, guidelines from the American Heart Association [26] do not include recommendations for cholesterol intake, but note that individuals should consume as little cholesterol as possible. French guidelines recommend limiting red meat and eggs to 500 g per week, but do not give guidelines for eggs or cholesterol specifically [27,28].

In view of inconsistencies in previous studies, we sought to investigate the associations between egg consumption, dietary cholesterol, and incident hypertension in a large prospective study of French women, and assessed these relations over a number of potential lifestyle and dietary factors.

## 2. Materials and Methods

### 2.1. Study Population

The Etude Epidémiologique de femmes de la Mutuelle Générale de l′Education (E3N) [29] is a French prospective cohort started in 1990 comprising 98,995 women aged 40–65 years at baseline and insured by the MGEN (Mutuelle Générale de l´Education Nationale), a health insurance plan for workers in the National Education System and their families. The objective of E3N was to study the main risk factors of cancer and chronic diseases. The E3N is the French component of the European Prospective Investigation into Cancer and Nutrition. The cohort received ethical approval from the French National Commission for Computerized Data and Individual Freedom (Commission Nationale Informatique et Libertés), and all participants in the study signed an informed consent.

Participants returned mailed questionnaires on lifestyle information and disease occurrence every 2 to 3 years. The average response rate at each questionnaire cycle was 83%, and the total loss to follow-up was 3%.

We included women who responded to a dietary questionnaire in 1993 (n = 74,522), and excluded women with prevalent hypertension, coronary disease or stroke (n = 26,974) before or at the 1993 questionnaire and those with unrealistic energy consumption (the 1st and 99th percentiles of the distribution in the population, n = 938). The final study population included 46,501 women. A flow diagram of exclusions (Appendix A), as well as a comparison between the excluded and included women is included in the supplementary material (Appendix A).

### 2.2. Assessment of Egg and Cholesterol Intake

In 1993, dietary data was collected using a two-part questionnaire detailing consumption of 208 food items which has been shown to be valid and reproducible [30]. Women were asked to answer questions about quantities and frequencies of consumption of food groups. Eleven possible responses were available, never or less than once a month; 1 to 3 times a month, and 1 to 7 times a week. A photo booklet was added to help estimate portion sizes [31]. From this questionnaire and using a detailed food composition table, mean daily intakes of energy (excluding energy from alcohol), alcohol, and nutrients including cholesterol were estimated.

Total egg consumption was based on the number and frequency of consumption of cooked eggs (i.e., poached or boiled eggs) or eggs from mixed dishes (i.e., omelettes). Egg consumption per week was estimated as the weekly intake in grams estimated from the questionnaire, divided by 60 g (the estimated weight of one egg in grams). The validity and reproducibility of the dietary questionnaire was previously assessed in a sample of 119 women who had completed two diet history questionnaires and twelve monthly 24-h dietary recalls over a one-year period [30]. The correlation coefficient between the dietary questionnaire and the 24-h recalls was 0.49 for eggs and 0.40 for dietary cholesterol. Cholesterol content for eggs was estimated as 380 mg/100 g of egg [32].

A second similar dietary questionnaire sent in 2005 allowed follow-up of dietary data. This questionnaire was completed by 82% of the study population. If a participant did not complete the second dietary questionnaire, the dietary data were imputed as the value from the first dietary questionnaire.

The outcome for this study was self-reported hypertension, validated through identification of an anti-hypertensive medication. Participants were asked to report whether they had hypertension at baseline (1993) and in each follow-up questionnaire (1994, 1997, 2000, 2002, 2005, 2008, 2011, and 2014), the date of diagnosis, and the use of antihypertensive treatments. The month and year of diagnosis were provided for most cases (69%). For individuals who were missing the month of diagnosis (14% of cases), it was imputed to June of the year of diagnosis. The median time between the date of diagnosis and the date of response to the first questionnaire after diagnosis was 12 months. Thus, for the cases with no year of diagnosis (n = 17%), we assigned it to be 12 months before they reported hypertension in a questionnaire. In 2004, a drug reimbursement database became available for 97.6% of participants. We used the self-reported date of diagnosis or the first date of drug reimbursement for antihypertensive medications (Anatomical Therapeutic Chemical Classification System codes C02, C03, C07, C08, and C09) whatever happened first, as the date of diagnosis for cases identified after 2004. Using the information of the MGEN health insurance plan drug claim database, we assessed the validity of self-reported hypertension within the E3N cohort, up to 2008. We compared hypertension self-report to antihypertensive drug reimbursement (any of the above specified codes). A positive predictive value of 82% was observed [33].

### 2.3. Assessment of Covariates

Family history of hypertension, education (elementary school education, high school education, and university education), and smoking (ever smoker, current smoker, or never smoker) were based on self-reports, and for treated diabetes and treated dyslipidaemia (i.e., use of lipid lowering medications) we used cases which had been validated through the use of the drug reimbursement database [34]. Age at menopause was determined on a combination of variables as previously reported [35]. The use of menopausal hormone therapy (MHT) was assessed at baseline, using a booklet containing photos of all types of oestrogens and progestogens. A Prudent/Mediterranean diet score was determined from dietary data using principal component analysis as previously described [36].

We assessed usual physical activity with a questionnaire in 1993 that included questions on weekly hours spent walking, cycling and performing light and heavy household chores, or recreational activities and sports considering the winter and summer seasons. It included questions on the time spent walking (to work, shopping, and leisure time), cycling (to work, shopping, and leisure time), housework, and sports activities (such as racket sports, swimming). Metabolic equivalents (METs) per week were estimated by multiplying the hourly average METs for each item based on values from the Compendium of Physical Activities [37] by the reported activity duration.

Self-reported height and weight were used to calculate body mass index (BMI), defined as weight (kg) divided by squared height (m^2^). In the cohort, self-reported anthropometry is considered reliable from a validation study [38].

Variables considered in the analyses, including egg consumption, diabetes, dyslipidaemia, physical activity, smoking, menopausal status, use of MHT, and BMI were considered at baseline and then updated using data from the 2005 questionnaire.

### 2.4. Statistical Analysis

Participants were split into groups depending on egg consumption as <1, 1–1.9, 2–2.9, 3–3.9, 4–6.9 and ≥7 eggs/week. Similarly, participants were grouped into quintiles of cholesterol intake defined by the population distributions. Time at entry was the age at the beginning of follow-up (1993), exit time was the age when participants were diagnosed with hypertension, died (dates of death were obtained from the participants’ medical insurance records), were lost to follow-up, or were censored at the end of the follow-up period (15 June 2014), whichever occurred first. Hazard ratios and 95% confidence intervals were estimated from Cox regression models with age as the time scale. P-values for trends were calculated using the median category value as a semi-continuous variable in the models.

A model was first assessed with time-updated egg consumption, and cholesterol consumption, and age as timeline (Model 0). Models were first adjusted for BMI (Model 1), then various known risk-factors for hypertension: total physical activity (MET-hours/week), total calories (KCAL), diabetes and dyslipidaemia status, menopausal status, use of MHT, family history of cardiovascular disease (yes/no), smoking (never, former, and current at baseline), education (no high school diploma, high school diploma) (Model 2), and dietary variables (salt, potassium, saturated fats, fruit and vegetables, alcohol, and cholesterol containing foods: unprocessed meat, processed meat, fish, shellfish) (Model 3); and in the case of cholesterol as exposure, Model 2 + intake of vegetables, fruits, salt, potassium and saturated fats (Model 3*). Finally, models were adjusted as model 2 plus a Mediterranean dietary pattern in place of other nutritional variables (Model 4). We considered Model 4 as the main model. As dyslipidaemia and diabetes may affect both exposure and outcome, models with and without these variables were assessed, but no large difference in estimates was obtained, thus the variables were retained. All covariates were updated at the 2005 time-point.

Subgroup analyses were performed for the following stratifications in order to assess the relations in groups associated with lifestyle factors and cholesterol intake: non-obese/excess weight, dyslipidaemia/diabetes at baseline, low reported processed meat intake/high reported processed meat intake, low/high Mediterranean diet score, current and ex-smokers/never smokers, and education level. A number of sensitivity analyses were performed. In order to account for reverse causation, we excluded cases diagnosed within five and then within 10 years.

Correlations between dietary variables were assessed using Spearmans correlation coefficient (r). Missing values occurred in less than 5%, and were imputed using the mean for continuous, or median for categorical variables. All statistical analyses used R version 3.5.1 (www.r-project.org) and the survival package, with an alpha of statistical significance equal to 0.05. Results from cox-models were interpreted as hazard ratio (HR) (95% confidence interval (CI)). The proportional hazards assumption was assessed using the cox.zph function in R.

## 3. Results

A total of 46,424 women were included in this study. After an average of 19.1 years of follow-up and 885,321 person years, 13,161 cases of incident hypertension were identified, corresponding to an incidence rate of 14.9 per 1000 person-years. At baseline, the mean (standard deviation) age was 50.1 (6.2) years, the mean BMI was 22.2 (2.8), and the mean physical activity was 54.4 (29.8) MET-hours/week.

At baseline, the mean egg consumption in the cohort was 2.7 (2.0) eggs/week, including 2.0 (1.5) eggs/week from omelettes and mixed dishes, and 0.8 (1.0) eggs/week from boiled eggs. The mean daily cholesterol intake was 366 (126) mg/day. Cholesterol from eggs constituted on average 22.4 (12.8)% of total dietary cholesterol at baseline. Compared to low consumption, high consumers of egg and cholesterol were younger and were less likely to be menopausal, more likely to smoke, and less likely to report treated dyslipidaemia (Table 1). Cholesterol consumption correlated strongly with intakes of eggs (r = 0.67) saturated fat (r = 0.72), processed meat (0.41), shellfish (r = 0.37), and liver (r = 0.34). Egg consumption correlated with intakes of shellfish (r = 0.42), processed meat (r = 0.23), Mediterranean diet score (r = 0.20), and saturated fat (r = 0.22).

In cox models using updated exposures (Table 2), the highest consumers of cholesterol were associated with a higher risk of hypertension than the lowest consumers (M4, HR_Q1–Q5_ = 1.22 [1.14: 1.30], *p* < 0.05). Associations were weakened, but remained significant when controlling on saturated fat intake (M3, HR_Q1–Q5_ = 1.10 [1.03: 1.18], *p* < 0.05). Similarly, egg consumption up to 7 eggs/week was associated with a higher risk of hypertension (M4, HR_4–7 eggs_ = 1.14 [1.06: 1.18]), whereas over 7 eggs/week was not (M4, HR_>7 eggs_ = 1.04 [0.96: 1.14]). Considering a mutually adjusted model for egg and cholesterol intake, associations were consistent for cholesterol (HR_Q1–Q5_ = 1.21 [1.12: 1.31], *p* < 0.05), but associations with eggs were null (HR_4–7 eggs_ = 1.04 [0.97: 1.12], *p* = 0.11) suggesting that cholesterol was driving the associations between egg consumption and hypertension.

Spline analysis (Figure 1) revealed a steep sharp dose-response relationship in low cholesterol consumers, which decreased in steepness at around 400 mg/day. For eggs, spline analysis showed a linear response from zero to around 7 eggs/week, after which a return to no extra risk was observed at around 10 eggs/week. 

### Subgroup Analyses

In analysis of groups we hypothesised as less likely to be confounded, a number of stratified models were assessed (Table 3 and Table 4). Results were similar amongst most subgroups of women. Considering egg intake, only amongst the high consumers of processed meat was there no clear evidence of an association between egg consumption and hypertension risk (p-interaction <0.005) (Table 3). When considering cholesterol, associations were not evident for high consumers of processed meat, and those with a low Mediterranean diet score (both p-interaction <0.005) (Table 4). Amongst smokers, associations were null for both egg and cholesterol intake, but the statistical test for interaction for this variable was not significant.

In sensitivity analyses excluding cases diagnosed within five and 10 years in order to account for reverse causation bias, associations were unchanged (data not tabulated).

## 4. Discussion

The results from this large prospective study suggest that high cholesterol and egg consumption is associated with a higher risk of hypertension. These results were independent of saturated fat, salt, processed meat/fish intakes, protective dietary factors such as vegetable and dietary fibre intake, and Mediterranean diet adherence. Associations between egg consumption and hypertension were null when adjusting for total cholesterol, indicating that cholesterol was driving the associations with eggs in this cohort. Using splines, we observed a non-linear response between cholesterol consumption and hypertension risk, with the steepest increase in low cholesterol consumers. For eggs, the risk increased linearly until around seven eggs per week, then decreased.

Few epidemiological studies have assessed the relationship between dietary cholesterol and blood pressure; however, they do find positive associations, which are in agreement with our results. Two previous studies involving American males have identified positive associations between cholesterol consumption and blood pressure increases [39,40]. The Multiple Risk Factor Intervention Trial [39] trial found that cholesterol intake was positively associated with blood pressure, even after adjustment for multiple dietary variables including saturated fat. The Chicago Western Electric study [40] confirmed these observations. Similarly, the multi-centre INTERMAP study reported positive associations between cholesterol intake and blood pressure in a multi-national cohort [24].

The observations regarding egg consumption agree with the observations from the Lipid and Blood Pressure Meta-analysis Collaboration (LBPMC) Group [25] (n = 23,524), who observed higher blood pressure in higher consumers of eggs, and a recent European study that identified associations between higher blood pressure and high consumers of eggs [7]. However, our observations disagree with four previous prospective investigations into egg consumption and hypertension [41,42,43,44] and one meta-analysis which reported lower associations of hypertension amongst egg consumers [10]. In the prospective CARDIA study [42] (n = 4304, cases = 997), higher egg intake was associated with lower risk of hypertension. An Iranian study [41] (n = 1708, cases = 144) observed a decreased incidence of hypertension in participants consuming upwards of around half an egg per day. Trends were similar in a Korean (n = 6792, cases = 1212) [44] and a Japanese study [43] (n = 3486, cases = 846). The Finnish KIHD study identified null associations between egg/cholesterol consumption [17], blood pressure, and stroke, but included only men, with a higher baseline cholesterol intake than our population. Eggs may vary in size and cholesterol content depending on geographic location, which could be a reason for differing results. It is likely that the overall diet, population characteristics, and regional differences in chronic disease epidemiology also affect the results.

Associations between dietary cholesterol and hypertension are possibly caused by higher levels of serum cholesterol in those with a high cholesterol intake. A recent meta-regression [22] using data solely from randomised trials indicates that a 100-mg/d increase in dietary cholesterol would result in a 4.52-mg/dL increase in circulating LDL-C, with the effect more pronounced in women. Higher serum cholesterol could be associated with increased blood pressure due to effects on nitric oxide (NO). Studies have shown that high cholesterol is strongly correlated with higher serum dimethylarginine (ADMA), and upregulates its synthesis in human endothelial cells [45]. ADMA inhibits NO production, a naturally derived substance that produces vasodilation and modulates blood pressure [46]. Inhibition of NO results in reduced dilatory capability of the blood vessels, endothelial dysfunction, and reduced salt sensitivity [47], leading to increased blood pressure [48] and arterial stiffening. Egg consumption has been associated with increases in both LDL-C and HDL-C [49,50,51], without significantly altering the ratio of HDL:LDL-C [49,50,51]. Increased HDL-C is implicated in the excretion of cholesterol via the liver in a process known as reverse cholesterol transport [52], and has various anti-oxidising effects, whereas high LDL-C is strongly associated with increased rates of CVD [53]. Lowering circulating cholesterol has been demonstrated to improve endothelial function [54], thus potentially explaining why increased cholesterol intake from sources such as eggs may have negative implications for vascular health, and incident hypertension [55].

In Asian cuisine, eggs are commonly incorporated into dishes, whereas in Europe, eggs are typically consumed with red or processed meat. Replacement effects could be in place for foods associated with hypertension in different regions, i.e., egg consumption was inversely related with salted seafood in the Korean study. Egg consumption is also difficult to compare between studies, for example, in the Japanese study, participants reported egg consumption as yes/no, whereas in the CARDIA study eggs were assessed as ‘times eaten per day’. In the Korean study, the median egg consumption was lower than in our cohort, 12.5 g per day, which we would have estimated as 1.2 eggs per week. Additionally, previous studies into egg consumption and hypertension have failed to estimate the dietary cholesterol intake of their participants, thus it is possible that associations could differ in populations consuming different amounts of dietary cholesterol or other nutrients.

Studies into cardiovascular mortality have reported increased risk associated with egg consumption [2,5,16]. A recent prospective analysis by Zhong et al. in the USA [2] identified positive associations between egg consumption, cholesterol intake (with a level of cholesterol intake in their multiple cohorts similar to ours), and incident CVD/all-cause mortality. Conversely, a recent Chinese study observed an inverse relationship between egg consumption and CVD [15]. In meta-analysis of egg consumption and CVD, inverse associations have been reported in Asian populations, but not in American or European populations [56].

In our study, the associations between cholesterol or egg intake and hypertension in subgroups were consistent. However, when considering those women who consumed over seven eggs per week, associations were null, perhaps due to the small sample size (~5% of the study population), or specific confounding with this group as they were significantly more likely to have attended university, and had a much higher Mediterranean diet score than the other groups of egg consumers. It is possible that egg intake this high represented a specific health-conscious behaviour in this population. Cholesterol and egg intake was associated with hypertension risk in those with a high Mediterranean diet score, suggesting that high cholesterol intake can have a negative impact on a high quality diet. In line with this, we observed weaker association between egg or cholesterol consumption and hypertension risk in high consumers of processed meat, a group who consumed more cholesterol, salt and saturated fats, and thus likely encountered no increased risk from egg-based or other sources of cholesterol.

## 5. Strengths and Limitations

The main strengths of this study are the time-updated cholesterol and egg data. The high number of egg consumers in this cohort, its prospective manner, large cohort and number of cases, limited loss to follow-up and detailed and validated dietary questionnaire, permitting us to assess updated coefficients in modelling, are also important factors. As with all observational studies, there are limiting factors. We relied on self-reported diagnosis, and prescription of anti-hypertensive drugs for the identification of hypertension cases. We observed a strong PPV of 82% for self-reported cases with drug reimbursements, but we could not identify undiagnosed cases. These cases should be randomly distributed, and would tend to attenuate associations.

The main weaknesses of the study are the self-reported diet, and the impossibility of claiming causality. The questionnaire, whilst validated using 24-h recalls, is subject to some variation. In this cohort, both eggs and cholesterol showed only modest recall correlation in 119 participants. It is difficult to rule out residual confounding due to dietary history, genetics, or lifestyle factors such as health awareness. Additionally, the E3N cohort is not representative of the French population, and is characterised by a high level of education, and a relatively low average BMI. However, we were able to adjust for the known risk-factors for hypertension onset, as well as various dietary factors associated with the disease such as fibre, salt, saturated fat and processed meat intake. We were also able to control for a Mediterranean dietary score, which has shown protective associations with various diseases in this cohort, the lack of which is a noted limitation of other studies in a recent meta-analysis. Associations were weakened, but still consistent when controlling for saturated fat. We performed various subgroup analyses to investigate the likelihood of residual confounding from smoking status, weight, social-economic status, diet, and treatments for dyslipidaemia and diabetes, although we may have lacked statistical power amongst those with diabetes.

It is important to consider how eggs are consumed; we observed the highest correlation between eggs and processed meat, and no correlation with vegetables or dietary fibre. People who consume eggs as an omelette with vegetables would likely be associated with a different overall diet pattern than eggs consumed with ham and other processed meats. We did not consider eggs from other sources, such as baked goods, due to the low amount of egg per serving size that would have been estimated, and we were also unable to determine if participants were consuming egg whites or yolks differently.

Egg consumption has been associated with lifestyle characteristics but this varies according to regions and demographics. In the E3N cohort who consumed on average 366 mg of cholesterol daily, a higher consumption of eggs was associated with a lower prevalence of smoking, slightly higher activity levels, and slightly lower education attainment. Regarding other dietary factors, high egg consumption was associated with increased processed meat consumption, but not with fruit and vegetable consumption; it also tended to be associated with a higher Mediterranean diet score. As the cohort consists of women, results may not be generalizable to men due to differences in lipid metabolism and protective effects from different hormones [57]; however, results in previous studies have been consistent across sex [42].

## 6. Conclusions

In conclusion, these results indicate that high cholesterol and egg consumption is associated with a higher risk of hypertension among French women. Evidence was found for non-linear relationships between the number of eggs consumed per week, cholesterol consumption, and the risk of hypertension.

## Figures and Tables

**Figure 1 nutrients-12-01350-f001:**
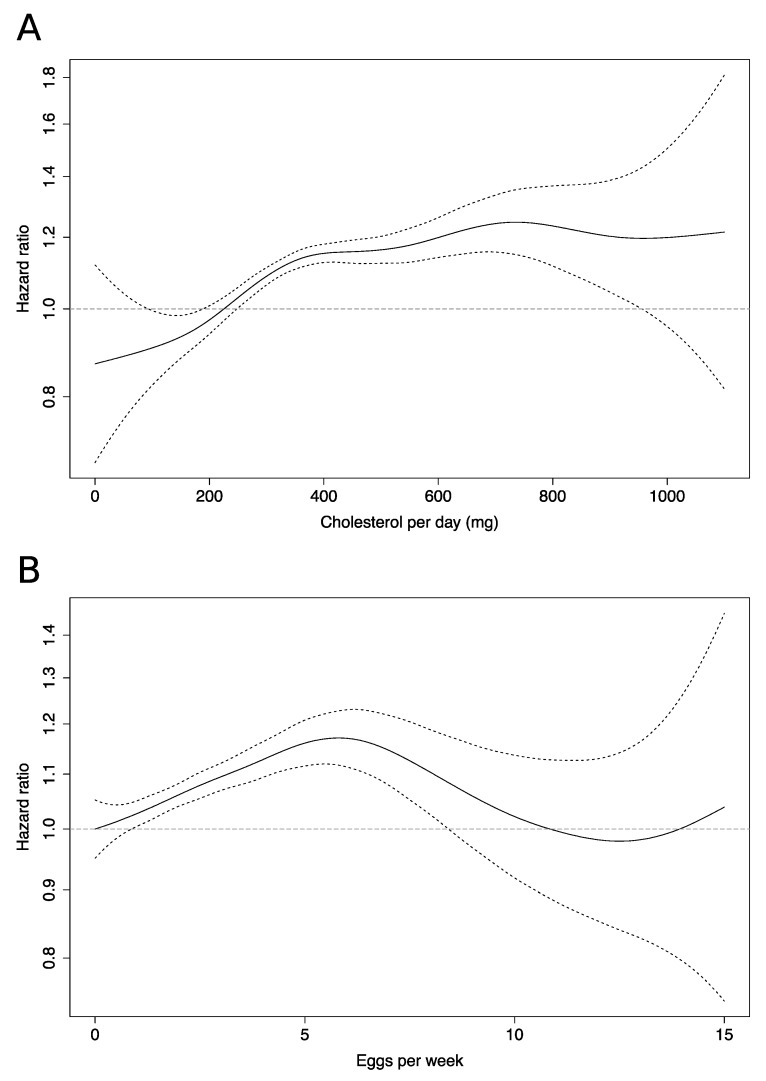
Spline regression curves detailing hypertension risk for associated with: (**A**) cholesterol consumption; (**B**) egg consumption. Dashed lines indicate 95% confidence interval. All models adjusted for BMI, total physical activity, smoking, family history of CVD, education level, menopausal status, use of MHT, dyslipidaemia, diabetes, intake of alcohol, total calorific intake, Mediterranean diet score and age as the timescale.

**Table 1 nutrients-12-01350-t001:** Participant demographics and dietary data, depending on egg and cholesterol consumption at baseline.

	Weekly Egg Intake (Eggs/Week)						Daily Cholesterol Intake (mg/Day)				
Variables (Mean, SD)	<1 (n = 8245)	1–1.9 (n = 11,189)	2–2.9 (n = 8927)	3–3.9 (n = 7303)	4–6.9 (n = 8307)	≥7 (n = 2453)	≤270 (n = 9826)	271–336 (n = 9825)	337–401 (n = 9825)	402–487 (n = 9825)	≥488 (n = 9823)
Median eggs/week	0.5 (0.3)	1.5 (0.3)	2.5 (0.3)	3.3 (0.3)	5.0 (0.8)	8.5 (2.7)	1.0 (0.9)	1.7 (1.2)	2.4 (1.4)	3.3 (1.7)	4.9 (3.1)
Median cholesterol (mg/day)	267 (93)	329 (95)	379 (99)	420 (102)	474 (116)	621 (154)	226 (42)	306 (19)	369 (19)	441 (25)	568 (105)
Age (years)	51.8 (6.7)	50.5 (6.3)	49.9 (6.1)	49.5 (6.0)	49.6 (5.8)	49.9 (6.1)	51.8 (6.7)	50.5 (6.3)	49.8 (6.1)	49.5 (6.0)	49.1 (5.8)
BMI (kg/m^2^)	21.8 (2.6)	22.1 (2.7)	22.2 (2.8)	22.4 (2.8)	22.7 (2.9)	22.9 (3.1)	21.8 (2.6)	22.1 (2.7)	22.2 (2.7)	22.4 (2.8)	22.8 (3.1)
Total physical activity (METS/week)	53.3 (29.7)	53.5 (29.2)	54.6 (29.8)	55.5 (30.2)	55.6 (30.8)	56.2 (31.1)	53.4 (29.9)	53.3 (29.3)	54.1 (29.2)	54.5 (29.6)	56.3 (31.0)
Diabetes (%)	0.4	0.4	0.6	0.6	0.6	0.7	0.5	0.5	0.5	0.5	0.6
Dyslipidaemia (%)	9.0	6.0	4.5	4.1	3.4	2.4	8.9	5.3	5.0	3.9	3.3
Family history CVD (%)	34.0	34.5	34.2	33.5	32.9	31.1	33.6	33.9	34.3	33.4	33.4
Prior cancer (%)	6.4	6.8	6.8	6.4	6.8	6.5	6.6	6.8	6.8	6.4	6.4
Menopausal at baseline (%)	50.2	43.2	40.3	38.4	40.1	42.1	52.5	44.4	40.5	38.3	36.5
Ever use of MHT amongst menopausal women (%)	44.3	46.7	46.9	47.2	46.3	45.5	42.6	46.9	47.7	47.9	46.9
High school education (%)	59.8	59.8	61.2	62.2	64.2	57.1	61.1	59.8	60.0	61.3	63.7
University education (%)	40.2	40.2	38.8	37.8	35.8	42.9	38.9	40.2	40.0	38.7	36.3
Smoking (Never/X/current) (%)	53.2/33.3/13.4	53.1/33.7/13.2	52.0/34.3/13.7	51.8/33.8/14.4	50.0/34.2/15.8	49.7/32.2/18.1	53.9/32.3/13.8	51.9/34.0/14.1	51.7/34.6/13.7	51.6/34.2/14.2	50.8/33.8/15.3
**Dietary variables (median, SD)**											
Total energy (Kcal)	1882 (489)	1995 (484)	2104 (502)	2237 (560)	2267 (548)	2375 (603)	1626 (357)	1885 (369)	2079 (385)	2284 (416)	2642 (508)
Salt (mg/day)	2478 (861)	2609 (836)	2760 (858)	2947 (924)	2987 (921)	3099 (1010)	2147 (706)	2463 (734)	2722 (755)	2983 (783)	3453 (918)
Potassium (mg/day)	3474 (993)	3601 (953)	3720 (991)	3963 (1061)	4012 (1086)	4195 (1212)	3225 (934)	3475 (914)	36845 (905)	3908 (961)	4344 (1058)
Saturated fats (g/day)	30 (12)	33 (12)	36 (12)	36 (12)	38 (13)	41 (15)	24 (7)	31 (7)	35 (8)	40 (10)	49 (13)
Dietary fibre (g/day)	23 (8)	23 (7)	24 (8)	25 (8)	25 (9)	26 (9)	22 (8)	23 (7)	24 (7)	25 (8)	27 (8)
Alcohol (g/day)	5 (13)	6 (12)	8 (13)	8 (14)	8 (15)	9 (16)	4 (11)	6 (13)	7 (13)	9 (14)	10 (16)
Meat (g/day)	76 (45)	81 (42)	81 (43)	80 (42)	74 (43)	69 (46)	58 (35)	73 (38)	82 (40)	86 (44)	91 (49)
Processed meat (sausage, ham, pate) (g/day)	10 (16)	13 (15)	16 (16)	20 (19)	21 (20)	23 (25)	9 (12)	12 (13)	16 (16)	19 (17)	26 (22)
Fish (g/day)	21 (23)	25 (20)	25 (20)	25 (21)	25 (23)	25 (26)	18 (20)	24 (20)	25 (21)	25 (22)	25 (25)
Shellfish/canned fish (g/day)	3 (9)	5 (10)	7 (10)	8 (11)	9 (12)	10 (15)	2 (7)	4 (8)	6 (9)	9 (11)	12 (15)
Vegetables (g/day)	143 (102)	143 (98)	143 (98)	157 (103)	151 (104)	171 (121)	143 (104)	143 (99)	143 (100)	148 (100)	157 (107)
Fruit (g/day)	231 (175)	228 (158)	227 (164)	219 (162)	229 (175)	233 (190)	237 (175)	231 (164)	226 (162)	219 (163)	219 (177)
Mediterranean diet score [−1, 1]	−0.34 (0.92)	−0.28 (0.88)	−0.19 (0.90)	−0.06 (0.93)	0.14 (1.00)	0.50 (1.15)	−0.37 (0.87)	−0.27 (0.87)	−0.17 (0.90)	−0.05 (0.95)	0.26 (1.09)

**Table 2 nutrients-12-01350-t002:** BMI adjusted and multivariate adjusted cox proportional hazard models for incident hypertension risk based on time varying distributions of egg consumption and cholesterol consumption, with follow-up from 1993–2014.

**Weekly Egg Intake**							
	**<1.0**	**1.0–1.9**	**2–2.9**	**3.0–3.9**	**4.0–6.9**	**≥** **7.0**	**P for trend**
Cases	2512	3144	2428	1959	2342	761	
Person years	179,297	218,736	165,256	129,013	145,639	49,387	
M0	ref	1.06 [1.01: 1.12]	1.10 [1.04: 1.17]	1.15 [1.09: 1.23]	1.22 [1.15: 1.29]	1.13 [1.05: 1.23]	<0.0005
M1	ref	1.03 [0.98: 1.09]	1.06 [1.00: 1.12]	1.10 [1.03: 1.16]	1.13 [1.07: 1.20]	1.04 [0.96: 1.12]	<0.0005
M2	ref	1.03 [0.98: 1.09]	1.06 [1.01: 1.13]	1.09 [1.03: 1.16]	1.11 [1.05: 1.17]	1.04 [0.96: 1.13]	<0.0005
M3	ref	1.01 [0.95: 1.06]	1.04 [0.98: 1.10]	1.07 [1.01: 1.13]	1.11 [1.05: 1.18]	1.00 [0.92: 1.09]	0.001
M4	ref	1.03 [0.98: 1.09]	1.07 [1.01: 1.13]	1.10 [1.04: 1.17]	1.14 [1.06: 1.18]	1.04 [0.96: 1.14]	<0.0005
**Quintiles of Cholesterol Intake (mg/day)**							
	**≤270**	**271–336**	**337–401**	**402–487**	**≥** **488**	**P for trend**	
Cases	2488	2614	2689	2607	2709		
Person years	176,926	177,661	177,362	177,884	175,639		
M0	ref	1.15 [1.09: 1.22]	1.19 [1.13: 1.26]	1.24 [1.18: 1.31]	1.29 [1.22: 1.36]	<0.0005	
M1	ref	1.12 [0.98: 1.10]	1.15 [1.02: 1.14]	1.17 [0.97: 1.09]	1.18 [0.99: 1.11]	<0.0005	
M2	ref	1.13 [1.07: 1.20]	1.17 [1.10: 1.24]	1.20 [1.13: 1.28]	1.22 [1.14: 1.30]	<0.0005	
M3*	ref	1.04 [0.98: 1.10]	1.06 [0.99: 1.12]	1.08 [1.02: 1.16]	1.10 [1.03: 1.18]	0.007	
M4	ref	1.13 [1.07: 1.20]	1.17 [1.02: 1.14]	1.20 [1.13: 1.28]	1.22 [1.14: 1.30]	<0.0005	

M0 non adjusted, with age as timeline; M1 adjusted for BMI; M2 adjusted for total physical activity, total calories, smoking, family history of CVD, education level, menopausal status, use of MHT, dyslipidaemia, and diabetes; M3 adjusted for M2 + intake of alcohol, unprocessed meat, processed meat, vegetables, fruits, shellfish, fish, salt, potassium and saturated fats; M3* adjusted for M2 + intake of vegetables, fruits, salt, potassium and saturated fats; M4 adjusted for M2 + Mediterranean diet score.

**Table 3 nutrients-12-01350-t003:** Subgroup analysis: stratified multivariate cox proportional hazard models for incident hypertension risk related to weekly egg consumption.

Weekly Egg Consumption								
	P for Interaction	<1.0	1.0–1.9	2.0–2.9	3.0–3.9	4.0–6.9	≥7.0	P for Trend
Normal weight (mean BMI = 21.4, n = 39,966)	0.07	ref	1.05 [0.96: 1.11]	1.12 [1.05: 1.19]	1.12 [1.05: 1.20]	1.18 [1.10: 1.26]	1.04 [0.94: 1.15]	<0.0005
Overweight/obese (mean BMI = 27.1, n = 6635)	-	ref	1.02 [0.90: 1.15]	0.94 [0.82: 1.07]	1.09 [0.95: 1.24]	1.09 [0.96: 1.24]	1.13 [0.96: 1.34]	0.04
No dyslipidaemia (n = 36,461)	0.11	ref	1.02 [0.98: 1.10]	1.07 [1.01: 1.14]	1.13 [1.06: 1.20]	1.17 [1.10 1.25]	1.06 [0.97: 1.16]	<0.0005
Dyslipidaemia (n = 9963)	-	ref	1.01 [0.89: 1.14]	1.05 [0.92: 1.20]	0.96 [0.86: 1.11]	0.98 [0.84: 1.14]	1.00 [0.79: 1.25]	0.70
No diabetes (n = 45,670)	0.69	ref	1.02 [0.97: 1.08]	1.07 [1.01: 1.13]	1.10 [1.04: 1.17]	1.14 [1.08: 1.21]	1.05 [0.97: 1.15]	<0.0005
Diabetes (n = 754)	-	ref	0.78 [0.46: 1.31]	0.86 [0.53: 1.41]	0.93 [0.59: 1.47]	0.90 [0.55: 1.53]	0.59 [0.31: 1.14]	0.42
High school education (n = 25,555)	0.82	ref	1.04 [0.98: 1.09]	1.07 [1.01: 1.13]	1.10 [1.04: 1.17]	1.15 [1.08: 1.22]	1.05 [0.97: 1.15]	<0.0005
University education (n = 18,065)	--	ref	0.98 [0.90: 1.07]	1.07 [0.98: 1.18]	1.04 [0.94: 1.15]	1.12 [1.02 1.24]	0.97 [0.84: 1.13]	0.03
Smokers (n = 6622)	0.30	ref	1.03 [0.89: 1.19]	1.05 [0.90: 1.22]	1.02 [0.86: 1.20]	1.02 [0.88: 1.18]	0.86 [0.69: 1.07]	0.56
Ex-smokers (n = 15,677)	-	ref	0.94 [0.85: 1.02]	0.99 [0.90: 1.09]	1.03 [0.93: 1.14]	1.09 [0.99: 1.21]	1.01 [0.87: 1.17]	0.24
Never smokers (n = 24,215)	-	ref	1.10 [1.02: 1.18]	1.12 [1.03: 1.21]	1.17 [1.07: 1.27]	1.18 [1.09: 1.29]	1.13 [1.00: 1.26]	0.01
Low processed meat (median = 7.9 g/day, n = 23,309)	<0.005	ref	1.04 [0.97: 1.12]	1.07 [0.99: 1.16]	1.16 [1.06: 1.26]	1.09 [1.00: 1.15]	0.99 [0.84: 1.15]	0.04
High processed meat (median = 25.9 g/day, n = 23,292)	-	ref	1.00 [0.93: 1.08]	1.02 [0.94: 1.10]	1.01 [0.92: 1.09]	1.10 [1.01: 1.19]	0.99 [0.89: 1.10]	0.04
High Mediterranean diet score (median = 0.5, n = 23,300)	0.38	ref	1.01 [0.93: 1.09]	1.05 [0.97: 1.15]	1.08 [0.99: 1.17]	1.14 [1.05: 1.23]	1.04 [0.93: 1.15]	0.002
Low Mediterranean diet score (median = −0.7, n = 23,301)	-	ref	1.06 [0.98: 1.13]	1.08 [1.00: 1.17]	1.13 [1.04: 1.23]	1.14 [1.05: 1.23]	1.07 [0.92: 1.23]	0.01

Models adjusted for BMI, total physical activity, smoking, family history of CVD, education level, menopausal status, use of MHT, dyslipidaemia, diabetes, intake of alcohol, total calorific intake, and Mediterranean diet score, with age as the timeline, minus the stratification variable.

**Table 4 nutrients-12-01350-t004:** Subgroup analysis: stratified multivariate cox proportional hazard models for incident hypertension risk related to daily cholesterol consumption.

Quintiles of Cholesterol Intake (mg/Day)							
	P for Interaction	≤270	271–336	337–401	402–487	≥488	P for Trend
Normal weight (mean BMI = 21.4, n = 39,966)	0.85	ref	1.16 [1.09: 1.24]	1.22 [1.15: 1.30]	1.25 [1.17: 1.34]	1.24 [1.15: 1.34]	<0.0005
Overweight/obese (mean BMI = 27.1, n = 6635)	-	ref	1.07 [0.93: 1.23]	1.05 [0.91: 1.21]	1.11 [0.97: 1.28]	1.21 [1.05: 1.40]	0.006
No dyslipidaemia (n = 36,461)	0.25	ref	1.15 [1.08: 1.23]	1.19 [1.12: 1.27]	1.23 [1.15: 1.32]	1.25 [1.16 1.34]	<0.0005
Dyslipidaemia (n = 9963)	-	ref	1.07 [0.94: 1.21]	1.10 [0.96: 1.26]	1.08 [0.93: 1.25]	1.09 [0.94: 1.28]	0.25
No diabetes (n = 45,670)	0.79	ref	1.13 [1.07: 1.20]	1.17 [1.11: 1.24]	1.21 [1.13: 1.28]	1.22 [1.14: 1.30]	< 0.05
Diabetes (n = 754)	-	ref	1.08 [0.66: 1.76]	0.94 [0.58: 1.54]	1.17 [1.73: 1.89]	1.01 [0.61: 1.66]	0.93
High school education (n = 25,555)	0.28	ref	1.14 [1.06: 1.23]	1.20 [1.11: 1.29]	1.22 [1.13: 1.32]	1.27 [1.17: 1.38]	<0.0005
University education (n = 18,065)	--	ref	1.12 [1.02: 1.23]	1.14 [1.04: 1.26]	1.15 [1.04: 1.27]	1.16 [1.04: 1.20]	0.02
Smokers (n = 6622)	0.35	ref	1.09 [0.93: 1.27]	1.02 [0.87: 1.19]	1.03 [0.88: 1.21]	1.04 [0.88: 1.23]	0.90
Ex-smokers (n = 15,677)	-	ref	1.16 [1.05: 1.28]	1.25 [1.13: 1.38]	1.26 [1.14: 1.40]	1.26 [1.12: 1.41]	<0.0005
Never smokers (n = 24,215)	-	ref	1.13 [1.04: 1.22]	1.17 [1.08: 1.26]	1.21 [1.12: 1.32]	1.24 [1.14: 1.36]	0.0005
Low processed meat (median = 7.9 g/day, n = 23,309)	<0.005	ref	1.17 [1.09: 1.26]	1.21 [1.12: 1.31]	1.21 [1.11: 1.32]	1.20 [1.08: 1.33]	<0.0005
High processed meat (median = 25.9 g/day, n = 23,292)	-	ref	1.03 [0.94: 1.13]	1.05 [0.96: 1.15]	1.08 [0.99: 1.18]	1.06 [0.97: 1.16]	0.26
High Mediterranean diet score (median = 0.5, n = 23,300)	<0.005	ref	1.21 [1.10: 1.32]	1.32 [1.21: 1.44]	1.35 [1.23: 1.48]	1.35 [1.24: 1.48]	<0.0005
Low Mediterranean diet score (median = −0.7, n = 23,301)	-	ref	1.07 [1.00: 1.16]	1.05 [0.97: 1.14]	1.09 [1.00: 1.18]	1.11 [1.01: 1.22]	0.03

Models adjusted for BMI, total physical activity, smoking, family history of CVD, education level, menopausal status, use of MHT, dyslipidaemia, diabetes, intake of alcohol, total calorific intake, and Mediterranean diet score, with age as the timeline, minus the stratification variable.

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
