# Peer review of "Cholesterol and Egg Intakes, and Risk of Hypertension in a Large Prospective Cohort of French Women"

_nutrients, 2020, doi:10.3390/nu12051350_

Round 1

Reviewer 1 Report

This study is very interesting to show that egg and cholesterol intake are associate with a higher risk of hypertension in French women using the data from E3N prospective cohort study.

  1. The relationship between egg/cholesterol intake and CVD is controversial, and egg/cholesterol intake influences blood cholesterol level more than blood pressure, which are both major risk factor for CVD. In introduction, the relationship between egg/cholesterol intake and hypercholesterolemia should be discussed.
  2. In method, authors stated that egg intake was unavailable at the second dietary questionnaire. Please report what % of population has missing data of egg consumption.
  3. In dietary questionnaire, egg consumption was surveyed as 0-1, 1-2 etc. If participant consume one egg/week, the participant checks 0-1 or 1-2?
  4. As authors mentioned, the results may not be generalizable to men. Please discuss whether the effect of egg/cholesterol intake on hypertension is different between men and women in the previous studies.
  5. Although confounding factors were adjusted, participants consumed more egg and cholesterol also consumed more processed meat and saturated fat but less fruit and vegetable. In addition, > 7 egg/week was negatively associated with risk of hypertension. The conclusion that dose-respondent relationship should be modified.
  6. In P3, line 134, what is MHT?
  7. In P6, line 202, Fig.1 is missing.
  8. Table 2, state actual P values, not like <0.05.

Author Response

This study is very interesting to show that egg and cholesterol intake are associate with a higher risk of hypertension in French women using the data from E3N prospective cohort study.

  1. The relationship between egg/cholesterol intake and CVD is controversial, and egg/cholesterol intake influences blood cholesterol level more than blood pressure, which are both major risk factor for CVD. In introduction, the relationship between egg/cholesterol intake and hypercholesterolemia should be discussed.

    Thank you for this suggestion. We have included in the introduction that dietary cholesterol can increase serum cholesterol, and in turn this can affect blood pressure (lines 57-60). We have also expanded the discussion to include these mechanistic points (lines 438-448).

  2. In method, authors stated that egg intake was unavailable at the second dietary questionnaire. Please report what % of population has missing data of egg consumption.

    Egg consumption was available for the majority of responders to the second dietary questionnaire. 18 % of the study population did not respond to the second dietary questionnaire, and had dietary data imputed from the first questionnaire. We have clarified this in the text (lines 134-137).
  3. In dietary questionnaire, egg consumption was surveyed as 0-1, 1-2 etc. If participant consume one egg/week, the participant checks 0-1 or 1-2?

    As stated in the methods, data was collected from dietary questionnaires as weekly egg intake, which was then used to assess intake in grams per week. In order to scale this to ‘eggs per week’, we divided the egg intake in gram by 60, the estimated weight of one egg.

    We then grouped participants in terms of their weekly egg consumption as less than 1, 1-1.9, 2-2.9… and so on. We have made this clearer in the manuscript and tables (lines 101, 178). We have made a similar change for cholesterol in the tables.
  4. As authors mentioned, the results may not be generalizable to men. Please discuss whether the effect of egg/cholesterol intake on hypertension is different between men and women in the previous studies.

    Thank you. We have included this in the discussion (line 575). Previous studies have not shown effect modification from sex.
  5. Although confounding factors were adjusted, participants consumed more egg and cholesterol also consumed more processed meat and saturated fat but less fruit and vegetable. In addition, > 7 egg/week was negatively associated with risk of hypertension. The conclusion that dose-respondent relationship should be modified.

    Thank you, we have modified these statements on the dose-response to state that the observation was linear until around 7 eggs / week, and then returned to non-significance as the HR was approximately 1 for the highest consumers.

  6. In P3, line 134, what is MHT?

    Menopausal hormone therapy, this has been clarified in the text on line 160. We have added details on the assessment of menopausal variables, as this was missing.
  7. In P6, line 202, Fig.1 is missing.

    Thank you this has been amended on line 307.
  8. Table 2, state actual P values, not like <0.05.

    The p-values have been updated, however as many are very low, we opted to set the lower limit for reporting at 0.0005.

Reviewer 2 Report

The validity of the dietary questionnaire is based on a moderate correlation with the results of 24 hour recalls in only 119 individuals. This limitation should be clearly stated.

For the reader a pathophysiologic hypothesis why cholesterol intake and hypertension might correlate might be of interest.

Author Response

The validity of the dietary questionnaire is based on a moderate correlation with the results of 24 hour recalls in only 119 individuals. This limitation should be clearly stated.

Thanks, we agree, and this has been included on lines 541-542.

For the reader a pathophysiologic hypothesis why cholesterol intake and hypertension might correlate might be of interest.

Thank you for this suggestion. We have included some discussion on the effect of dietary cholesterol on serum cholesterol, and in turn on blood pressure regulation on lines 438-448.
